# ROTATIONAL UNIT OF MEMORY

## ABSTRACT

The concepts of unitary evolution matrices and associative memory have boosted the field of Recurrent Neural Networks (RNN) to state-of-the-art performance in a variety of sequential tasks. However, RNN still have a limited capacity to manipulate long-term memory. To bypass this weakness the most successful applications of RNN use external techniques such as attention mechanisms. In this paper we propose a novel RNN model that unifies the state-of-the-art approaches: Rotational Unit of Memory (RUM). The core of RUM is its rotational operation, which is, naturally, a unitary matrix, providing architectures with the power to learn long-term dependencies by overcoming the vanishing and exploding gradients problem. Moreover, the rotational unit also serves as associative memory. We evaluate our model on synthetic memorization, question answering and language modeling tasks. RUM learns the Copying Memory task completely and improves the state-of-the-art result in the Recall task. RUM's performance in the bAbI Question Answering task is comparable to that of models with attention mechanism. We also improve the state-of-the-art result by 0.001 to 1.189 bits-per-character (BPC) test loss in the Character Level Penn Treebank (PTB) task. Moreover, our models achieve 0.002 BPC improvement to the validation too, which is to signify the applications of RUM to real-world sequential data. The universality of our construction, at the core of RNN, establishes RUM as a promising approach to language modeling, speech recognition and machine translation.

## 1 INTRODUCTION

Recurrent neural networks are widely used in a variety of machine learning applications such as language modeling (Graves et al. (2014)), machine translation (Cho et al. (2014)) and speech recognition (Hinton et al. (2012)). Their flexibility of taking inputs of dynamic length makes RNN particularly useful for these tasks. However, the traditional RNN models such as Long Short-Term Memory (LSTM, Hochreiter & Schmidhuber (1997)) and Gated Recurrent Unit (GRU, Cho et al. (2014)) exhibit some weaknesses that prevent them from achieving human level performance: 1) limited memory–they can only remember a hidden state, which usually occupies a small part of a model; 2) gradient vanishing/explosion (Bengio et al. (1994)) during training–trained with back-propagation through time the models fail to learn long-term dependencies.

Several ways to address those problems are known. One solution is to use soft and local attention mechanisms (Cho et al. (2014)), which is crucial for most modern applications of RNN. Nevertheless, researchers are still interested in improving basic RNN cell models to process sequential data better. Numerous works (Graves et al. (2014); Ba et al. (2016a)) use associative memory to span a large memory space. For example, a practical way to implement associative memory is to set weight matrices as trainable structures that change according to input instances for training. Furthermore, the recent concept of unitary or orthogonal evolution matrices (Arjovsky et al. (2016); Jing et al. (2017b)) also provides a theoretical and empirical solution to the problem of memorizing long-term dependencies.

Here, we propose a novel RNN cell that resolves simultaneously those weaknesses of basic RNN. The Rotational Unit of Memory is a modified gated model whose rotational operation acts as associative memory and is strictly an orthogonal matrix.

We tested our model on several benchmarks. RUM is able to solve the synthetic Copying Memory task while traditional LSTM and GRU fail. For synthetic Recall task, RUM exhibits a stronger

ability to remember sequences, hence outperforming state-of-the-art RNN models such as Fast-weight RNN (Ba et al. (2016a)) and WeiNet (Zhang & Zhou (2017)). By using RUM we achieve the state-of-the-art result in the real-world Character Level Penn Treebank task. RUM also outperforms all basic RNN models in the bAbI question answering task. This performance is competitive with that of memory networks, which take advantage of attention mechanisms.

Our contributions are as follows:

1. We develop the concept of the Rotational Unit that combines the memorization advantage of unitary/orthogonal matrices with the dynamic structure of associative memory;

2. The Rotational Unit of Memory serves as the first phase-encoded model for Recurrent Neural Networks, which improves the state-of-the-art performance of the current frontier of models in a diverse collection of sequential task.

## 2 Motivation and Related Work

### 2.1 Unitary Approach

The problem of the gradient vanishing and exploding problem is well-known to obstruct the learning of long-term dependencies (Bengio et al. (1994)).

We will give a brief mathematical motivation of the problem. Let's assume the cost function is $C$. In order to evaluate $\partial C / \partial \mathrm{W}^{ij}$, one computes the derivative gradient using the chain rule:

$$\frac{\partial C}{\partial \mathbf{h}^{(t)}} = \frac{\partial C}{\partial \mathbf{h}^{(T)}} \frac{\partial \mathbf{h}^{(T)}}{\partial \mathbf{h}^{(t)}} = \frac{\partial C}{\partial \mathbf{h}^{(T)}} \prod_{k=t}^{T-1} \frac{\partial \mathbf{h}^{(k+1)}}{\partial \mathbf{h}^{(k)}} = \frac{\partial C}{\partial \mathbf{h}^{(T)}} \prod_{k=t}^{T-1} \mathrm{D}^{(k)} \mathrm{W},$$

where $\mathrm{D}^{(k)} = \mathrm{diag}\{\sigma'(\mathrm{W}\mathbf{x}^{(k)} + \mathrm{A}\mathbf{h}^{(k-1)} + \mathbf{b})\}$ is the Jacobian matrix of the point-wise non-linearity. As long as the eigenvalues of $\mathrm{D}^{(k)}$ are of order unity, then if W has eigenvalues $\lambda_i \gg 1$, they will cause gradient explosion $\left|\partial C / \partial \mathbf{h}^{(T)}\right| \to \infty$, while if W has eigenvalues $\lambda_i \ll 1$, they can cause gradient vanishing, $\left|\partial C / \partial \mathbf{h}^{(T)}\right| \to 0$. Either situation hampers the efficiency of RNN.

LSTM is designed to solve this problem, but gradient clipping (Pascanu et al. (2012)) is still required for training. Recently, by restraining the hidden-to-hidden matrix to be orthogonal or unitary, many models have overcome the problem of exploding and vanishing gradients. Theoretically, unitary and orthogonal matrices will keep the norm of the gradient because the absolute value of their eigenvalues equals one.

Several approaches have successfully developed the applications of unitary and orthogonal matrix to recurrent neural networks. Arjovsky et al. (2016); Jing et al. (2017b) use parameterizations to form the unitary spaces. Wisdom et al. (2016) applies gradient projection onto a unitary manifold. Vorontsov et al. (2017) uses penalty terms as a regularization to restrain matrices to be unitary, hence accessing long-term memorization.

Only learning long-term dependencies is not sufficient for a powerful RNN. Jing et al. (2017a) finds that the combination of unitary/orthogonal matrices with a gated mechanism improves the performance of RNN because of the benefits of a forgetting ability. Jing et al. (2017a) also points out the optimal way of such a unitary/gated combination: the unitary/orthogonal matrix should appear before the reset gate, which can then be followed by a modReLU activation. In RUM we implement an orthogonal operation in the same place, but the construction of that matrix is completely different: instead of parameterizing the kernel, we encode a natural rotation, generated by the inputs and the hidden state.

### 2.2 Associative Memory Approach

Limited memory in RNN is truly a shortage. Adding an external associative memory is a natural solution. For instance, the Neural Turing Machine (Graves et al. (2014)) and many other models have shown the power of using this technique. While it expands the accessible memory space, the technique significantly increases the size of the model, therefore making the process of learning so many parameters harder.

Now, we will briefly describe the concept of associative memory. In basic RNN, $\mathbf{h}_t = \sigma(W\mathbf{x}_t + A\mathbf{h}_{t-1} + \mathbf{b})$ where $\mathbf{h}_t$ is the hidden state at time step $t$ and $\mathbf{x}$ is the input data at each step. Here $W$ and $A$ are trainable parameters that are fixed in the model. A recent approach replaces $A$ with a dynamic $A_t$ (as a function of time) so that this matrix can serve as a memory state. Thus, the memory size increases from $O(N_h)$ to $O(N_h^2)$, where $N_h$ is the hidden size. In particular, $A_t$ is determined by $A_{t-1}$, $\mathbf{h}_{t-1}$ and $\mathbf{x}_t$ which can be a part of a multi-layer or a Hopfiled net. By treating the RNN weights as memory determined by the current input data, a larger memory size is provided and less trainable parameters are required. This significantly increases the memorization ability of RNN. Our model also falls into this category of associative memory through its rotational design of an orthogonal $A_t$ matrix.

## 2.3 CAPSULE REPRESENTATION APPROACH

Recently, Sabour et al. (2017) proposed a novel neural network architecture that uses vectors instead of conventional single neurons to represent concepts in hidden states. These vectors are called *capsules*. Special connections are also designed to connect capsules through a process, called dynamic routing. This work shows promising performance of phase-encoded models in Convolutional Neural Networks. The Rotational Unit of Memory model, which we introduce below, serves as the first successful phase-encoded model in the RNN domain. We give a detailed comparison of these two models in section 5.3.

## 3 METHODS

Rotations are well-studied mathematical structures that have various fundamental applications in the theory of Lie groups (Artin (2011); Hall (2015)), quantum physics (Sakurai & Napolitano (2010)), etc. In computer vision (Shapiro & Stockman (2001)) the position and orientation of an object form a *pose*, which contains valuable information about the object. A feasible way of estimating poses is through rotational matrices and quaternions (Katz (2001); Kuipers (2002)).

The conventional way of representing memory in RNNs is by encoding the information in a *hidden state*, which is a vector of a certain finite dimension $N$. To the best of our knowledge, the frontier of RNN models utilizes mostly the norm of the elements of the hidden state during the learning process. Experiments and theory point, however, that representational advantages can be achieved, by using capsules as vectors in the Euclidean $\mathbb{R}^N$ space and thereby allowing the model to manipulate the pose of these capsules (Sabour et al. (2017)).

Here, we equip the hidden state in an RNN with a pose by viewing it as a vector with position and orientation in $\mathbb{R}^N$. We then propose an efficient method for manipulating the pose of the orientation by the means of rotations in an $N$-dimensional space. Our particular parameterization for the rotation is a natural way to define a differentiable orthogonal operation within the RNN cell. [1]

For the remainder of this section we suggest ways of engineering models that incorporate rotations as units of memory. In the following discussion $N_x$ is the input size and $N_h$ is the hidden size.

## 3.1 THE OPERATION Rotation

The operation Rotation is an efficient encoder of an orthogonal operation, which acts as a unit of memory. Rotation computes an orthogonal operator $R(\mathbf{a}, \mathbf{b})$ in $\mathbb{R}^{N_h \times N_h}$ that represents the rotation between two non-collinear vectors $\mathbf{a}$ and $\mathbf{b}$ in the two-dimensional subspace $\mathrm{span}(\mathbf{a}, \mathbf{b})$ of the Euclidean space $\mathbb{R}^{N_h}$ with distance $\|\cdot\|$. As a consequence, $R$ can act as a kernel on a hidden state $\mathbf{h}$. More formally, what we propose is a function

$$\text{Rotation} \colon \mathbb{R}^{N_h} \times \mathbb{R}^{N_h} \to \mathbb{R}^{N_h \times N_h},$$

---

[1] Other ways to extract an orthogonal operation from elements in the RNN cell are still possible. Some approaches are as follows: 1. Use a skew-symmetric matrix $A$ to define the orthogonal operator $e^A$; 2. Use a permutation operator. However, those constructions are difficult to implement and do not offer a natural intuition about encoding memory. We recognize that other constructions are also feasible and potentially interesting for research.

such that after ortho-normalizing $\mathbf{a}$ and $\mathbf{b}$ to

$$\mathbf{u}_a = \frac{\mathbf{a}}{\|\mathbf{a}\|} \quad \text{and} \quad \mathbf{u}_b = \frac{\mathbf{b} - (\mathbf{u}_a \cdot \mathbf{b}) \cdot \mathbf{u}_a}{\|\mathbf{b} - (\mathbf{u}_a \cdot \mathbf{b}) \cdot \mathbf{u}_a\|}, \quad \text{for which} \quad \theta = \arccos \frac{\mathbf{u}_a \cdot \mathbf{u}_b}{\|\mathbf{u}_a\| \|\mathbf{u}_b\|},$$

we encode the following matrix in $\mathbb{R}^{N_h} \times \mathbb{R}^{N_h}$

$$R(\mathbf{a}, \mathbf{b}) = \left[ \mathbf{1} - \mathbf{u}_a^T \cdot \mathbf{u}_a - \mathbf{u}_b^T \cdot \mathbf{u}_b \right] + (\mathbf{u}_a, \mathbf{u}_b)^T \cdot \tilde{R}(\theta) \cdot (\mathbf{u}_a, \mathbf{u}_b) . \tag{1}$$

Figure 1 (a) demonstrates the projection to the plane $\mathrm{span}(\mathbf{a}, \mathbf{b})$ in the brackets of equation (1). The mini-rotation in this space is $\tilde{R}(\theta) = \begin{pmatrix} \cos\theta & -\sin\theta \\ \sin\theta & \cos\theta \end{pmatrix}$. Hence, $\mathrm{Rotation}(\mathbf{a}, \mathbf{b}) \equiv R(\mathbf{a}, \mathbf{b})$.

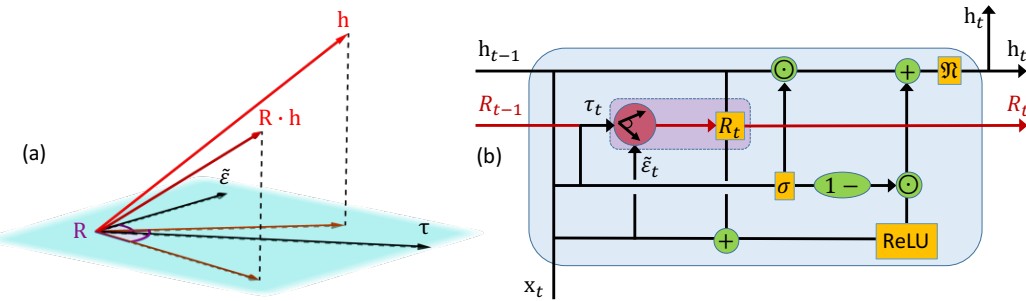

Figure 1: Rotation **is a universal differentiable operation that enables the advantages of the RUM architecture.** (a) The rotation $R(\mathbf{a}, \mathbf{b})$ in the plane defined by $\mathbf{a} = \tilde{\varepsilon}$ and $\mathbf{b} = \boldsymbol{\tau}$ acts on the hidden state $\mathbf{h}$. (b) The RUM cell, in which Rotation encodes the kernel $R$. The matrix $R_t$ acts on $\mathbf{h}_{t-1}$ and thus keeps the norm of the hidden state.

A practical advantage of Rotation is that it is both orthogonal and differentiable. On one hand, it is a composition of differentiable sub-operations, which enables learning via backpropagation. On the other hand, it preserves the norm of the hidden state, hence it can yield more stable gradients. We were motivated to find differentiable implementations of unitary (orthogonal in particular) operations in existing toolkits for deep learning. Our conclusion is that Rotation can be implemented in various frameworks that are utilized for RNN and other deep learning architectures. Indeed, Rotation is not constrained to parameterize a unitary structure, but instead it produces an orthogonal matrix from simple components in the cell, which makes it useful for experimentation.

We implement Rotation together with its action on a hidden state efficiently. We do not need to compute the matrix $R_t$ before we rotate. Instead we can directly apply the RHS of equation (1) to the hidden state. Hence, the memory complexity of our algorithm is $O(N_b \cdot N_h)$, which is determined by the RHS of (1). Note that we only use two trainable vectors in $\mathbb{R}^{N_h}$ to generate orthogonal weights in $\mathbb{R}^{N_h \times N_h}$, which means the model has $O(N_h^2)$ degrees of freedom for a single unit of memory. Likewise, the time complexity is $O(N_b \cdot N_h^2)$. Thus, Rotation is a universal operation that enables implementations suitable to any neural network model with backpropagation.

## 3.2 THE RUM ARCHITECTURE

We propose the Recurrent Unit of Memory as the first example of an application of Rotation to a recurrent cell. Figure 1 (b) is a sketch of the connections in the cell. RUM consists of an update gate $\mathbf{u} \in \mathbb{R}^{N_h}$ that has the same function as in GRU. Instead of a reset gate, however, the model learns a memory *target* variable $\boldsymbol{\tau} \in \mathbb{R}^{N_h}$. RUM also learns to embed the input vector $\mathbf{x} \in \mathbb{R}^{N_x}$ into $\mathbb{R}^{N_h}$ to yield $\tilde{\varepsilon} \in \mathbb{R}^{N_h}$. Hence Rotation encodes the rotation between the embedded input and the target, which is accumulated to the associative memory unit $R_t \in \mathbb{R}^{N_h \times N_h}$ (originally initialized to the identity matrix). Here $\lambda$ is a non-negative integer that is a hyper-parameter of the model. From here, the orthogonal $R_t$ acts on the state $\mathbf{h}$ to produce an evolved hidden state $\tilde{\mathbf{h}}$. Finally RUM obtains the

new hidden state via $\mathbf{u}$, just as in GRU. The RUM equations are as follows

$$\boldsymbol{\tau}_t = W_{xh}^{\tau} \cdot \mathbf{x}_t + W_{hh}^{\tau} \cdot \mathbf{h}_{t-1} + \mathbf{b}_t^{\tau} \qquad \text{memory target;}$$

$$\mathbf{u}_t' = W_{xh}^{u'} \cdot \mathbf{x}_t + W_{hh}^{u'} \cdot \mathbf{h}_{t-1} + \mathbf{b}_t^{u'} \qquad \text{initial update gate;}$$

$$\mathbf{u}_t = \mathsf{sigmoid}(\mathbf{u}_t') \qquad \sigma \text{ activation of the update gate;}$$

$$\tilde{\boldsymbol{\varepsilon}}_t = \tilde{W}_{xh} \cdot \mathbf{x}_t + \tilde{\mathbf{b}}_t \qquad \text{embedded input for Rotation;}$$

$$R_t = (R_{t-1})^{\lambda} \cdot \mathsf{Rotation}(\tilde{\boldsymbol{\varepsilon}}_t, \boldsymbol{\tau}_t) \qquad \text{rotational associative memory;}$$

$$\tilde{\mathbf{h}}_t = \mathsf{ReLU}\,(\tilde{\boldsymbol{\varepsilon}}_t + R_t \cdot \mathbf{h}_{t-1}) \qquad \text{unbounded evolution of hidden state;}$$

$$\mathbf{h}_t' = \mathbf{u}_t \odot \mathbf{h}_{t-1} + (1 - \mathbf{u}_t) \odot \tilde{\mathbf{h}}_t \qquad \text{hidden state before time normalization } \mathfrak{N};$$

$$\mathbf{h_t} = \eta \frac{\mathbf{h_t}}{\|\mathbf{h_t}\|} \qquad \text{new hidden state, with norm } \eta.$$

We have introduced time subscripts to demonstrate the recurrence relations. The kernels have dimensions given by $W_{xh}^{\tau}, W_{xh}^{u'} \in \mathbb{R}^{N_x \times N_h}$, and $W_{hh}^{\tau}, W_{hh}^{u'} \in \mathbb{R}^{N_h \times N_h}$, and $\tilde{W}_{xh} \in \mathbb{R}^{N_x \times N_h}$. The biases are variables, given by $\mathbf{b}_t^{\tau}, \mathbf{b}_t^{u'}, \tilde{\mathbf{b}}_t \in \mathbb{R}^{N_h}$. The norm $\eta$ is a scalar hyper-parameter of the RUM model.

The orthogonal matrix $R(\tilde{\varepsilon}_t, \boldsymbol{\tau})$ conceptually takes the place of a kernel acting on the hidden state in GRU. This is the most efficient place to introduce an orthogonal operation, as the Gated Orthogonal Recurrent Unit (GORU, Jing et al. (2017a)) experiments suggest. The difference with the GORU cell is that GORU parameterizes and learns the kernel as an orthogonal matrix, while RUM does not parameterize the rotation $R$. Instead, RUM learns $\boldsymbol{\tau}$, which together with $\mathbf{x}$, determines $R$. The orthogonal matrix keeps the norm of the vectors, so we experiment with a ReLU activation instead of the conventional `tanh` in gated mechanisms.

Even though $R$ is an orthogonal element of RUM, the norm of $\mathbf{h}_t'$ is not stable because of the ReLU activation. Therefore, we suggest normalizing the hidden state $\mathbf{h_t}$ to a have norm $\eta$. We call this technique *time normalization* as we usually feed mini-batches to the RNN during learning that have the shape $(N_b, N_T)$, where $N_b$ is the size of the batch and $N_T$ is the length of the sequence that we feed in. Time normalization happens along the sequence dimension as opposed to the batch dimension in batch normalization. Choosing appropriate $\eta$ for the RUM model stabilizes learning and ensures the eigenvalues of the kernels are bounded from above. This in turn means that the smaller $\eta$ is, the more we reduce the effect of exploding gradients.

Finally, even though RUM uses an update gate, it is not a standard gated mechanism, as it does not have a reset gate. Instead we suggest utilizing additional memory via the target vector $\boldsymbol{\tau}$. By feeding inputs to RUM, $\boldsymbol{\tau}$ adapts to encode rotations, which align the hidden states in desired locations in $\mathbb{R}^{N_h}$, without changing the norm of $\mathbf{h}$. We believe that the unit of memory $R_t$ gives advantage to RUM over other gated mechanisms, such as LSTM and GRU.

## 4 EXPERIMENTS

Firstly, we test RUM's memorization capacity on the Copying Memory Task. Secondly, we signify the superiority of RUM by obtaining a state-of-the-art result in the Associative Recall Task. Thirdly, we show that even without external memory, RUM achieves comparable to state-of-the-art results in the bAbI Question Answering data set. Finally, we utilize RUM's rotational memory to reach 1.189 BPC in the Character Level Penn Treebank.

We experiment with $\lambda = 0$ RUM and $\lambda = 1$ RUM, the latter model corresponding to tuning in the rotational associative memory.

### 4.1 COPYING MEMORY TASK

A standard way to evaluate the memory capacity of a neural network is to test its performance in the Copying Memory Task (Hochreiter & Schmidhuber (1997), Henaff et al. (2016) Arjovsky et al. (2016)). We follow the setup in Jing et al. (2017b). The objective of the RNN is to remember (copy) information received $T$ time steps earlier (see section A for details about the data).

Our results in this task demonstrate: 1. RUM utilizes a different representation of memory that outperforms those of LSTM and GRU; 2. RUM solves the task completely, despite its update gate, which does not allow all of the information encoded in the hidden stay to pass through. The only other gated RNN model successful at copying is GORU. Figure 2 reveals that LSTM and GRU hit a predictable baseline, which is equivalent to random guessing. RUM falls bellow the baseline, and subsequently learns the task by achieving zero loss after a few thousands iterations. The spikes on the learning curves for RUM are arising from the fact that we are using a ReLU activation for RUM without gradient clipping.

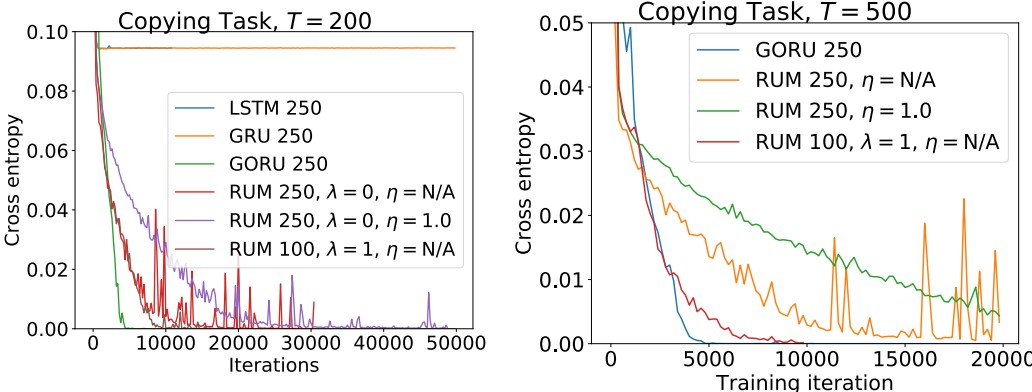

Figure 2: **The orthogonal operation** Rotation **enables RUM to solve the Copying Memory Task**. The delay times are $200$, $500$ and $1000$. For all models $N_h = 250$ except for the RUM models with $\lambda = 1$, for which $N_h = 100$. For the training of all models we use RMSProp optimization with a learning rate of $0.001$ and a decay rate of $0.9$; the batch size $N_b$ is $128$.

With the help of figure 2 we will explain how the additional hyper-parameters for RUM affect its training. We observe that when we remove the normalization ($\eta = \text{N/A}$) then RUM learns more quickly than the case of requiring a norm $\eta = 1.0$. At the same time, though, the training entails more fluctuations. Hence we believe that choosing a finite $\eta$ to normalize the hidden state is an important tool for stable learning. Moreover, it is necessary for the NLP task in this paper (see section 4.4): for our character level predictions we use large hidden sizes, which if left unnormalized, can make the cross entropy loss blow up.

We also observe the benefits of tuning in the associative rotational memory. Indeed, a $\lambda = 1$ RUM has a smaller hidden size, $N_h = 100$, yet it learns much more quickly than a $\lambda = 0$ RUM. It is possible that the accumulation of phase via $\lambda = 1$ to enable faster long-term dependence learning than the $\lambda = 0$ case. Either way, both models overcome the vanishing/exploding gradients, and eventually learn the task completely.

## 4.2 Associative Recall Task

Another important synthetic task to test the memory ability of recurrent neural network is the Associative Recall. This task requires RNN to remember the whole sequence of the data and perform extra logic on the sequence.

We follow the same setting as in Ba et al. (2016a) and Zhang & Zhou (2017) and modify the original task so that it can test for longer sequences. In detail, the RNN is fed into a sequence of characters, e.g. "a1s2d3f4g5??d". The RNN is supposed to output the character based on the "key" which is located at the end of the sequence. The RNN needs to look back into the sequence and find the "key" and then to retrieve the next character. In this example, the correct answer is "3". See section B for further details about the data.

In this experiment, we compare RUM to an LSTM, , a Fast-weight RNN (Ba et al. (2016a)) and a recent successful RNN WeiNet (Zhang & Zhou (2017)). All the models have the same hidden state $N_h = 50$ for different lengths $T$. We use a batch size 128. The optimizer is RMSProp with a learning rate 0.001. We find that LSTM fails to learn the task, because of its lack of sufficient

memory capacity. NTM and Fast-weight RNN fail longer tasks, which means they cannot learn to manipulate their memory efficiently. Table 1 gives a numerical summary of the results and figure 4, in the appendix, compares graphically RUM to LSTM.

| Model | Length $T = 30$ | Length $T = 50$ | # Parameters |
|---|---|---|---|
| LSTM | 25.6% | 20.5% | 17k |
| GORU | 21.8% | 18.9 % | 13k |
| FW-LN (Ba et al. (2016a) | 100% | 20.8% | 9k |
| WeiNet (Zhang & Zhou (2017)) | 100% | 100% | 22k |
| RUM (ours) | 100% | 100% | 13k |

Table 1: Comparison of the models on convergence validation accuracy. Only RUM and the recent WeiNet are able to successfully solve the $T = 50$ Associative Recall task with a hidden state of 50. RUM has significantly less parameters.

## 4.3 QUESTION ANSWERING

Question answering remains one of the most important applicable tasks in NLP. Almost all state-of-the-art performance is achieved by the means of attention mechanisms. Few works have been done to improve the performance by developing stronger RNN. Here, we tested RUM on the bAbI Question Answering data set (Weston et al. (2015)) to demonstrate its ability to memorize and reason without any attention. In this task, we train 20 sub-tasks jointly for each model, using a 10k training sets. See section C for detailed experimental settings and results on each sub-task.

We compare our model with several baselines: a simple LSTM, an End-to-end Memory Network (Sukhbaatar et al. (2015)) and a GORU. We find that RUM outperforms significantly LSTM and GORU and achieves competitive result with those of MemN2N, which has an attention mechanism. We summarize the results in Table 2. We emphasize that for some sub-tasks in the table, which require large memory, RUM outperforms models with attention mechanisms (MemN2N).

| Model | Test Accuracy (%) |
|---|---|
| LSTM (Weston et al. (2015)) | 49 |
| GORU (Jing et al. (2017a)) | 60 |
| MemN2N (Sukhbaatar et al. (2015)) | 86 |
| RUM (ours) | **73.2** |

Table 2: Question Answering task on bAbI dataset. Test accuracy (%) on LSTM, MemN2N, GORU and RUM. RUM outperforms LSTM/GORU and is outperformed only by MemN2N, which uses an attention mechanism.

## 4.4 CHARACTER LEVEL LANGUAGE MODELING

The Rotational Unit of Memory is a natural architecture that can learn long-term structure in data while avoiding significant overfitting. Perhaps, the best way to demonstrate this unique property, among other RNN models, is to test RUM on real world character level NLP tasks.

### 4.4.1 PENN TREEBANK CORPUS DATA SET

The corpus is a collection of articles in *The Wall Street Journal* (Marcus et al. (1993)). The text is in English and its vocabulary consists of 10000 words. We split the data into *train*, *validation* and *test* sets according to Mikolov et al. (2012). We train by feeding mini-batches of size $N_b$ that consist of sequences of $T$ consecutive characters.

We incorporate RUM into the state-of-the-art high-level model: Fast-Slow RNN (FS-RNN, Mujika et al. (2017)). The FS-RNN-$k$ architecture consists of two hierarchical layers: one of them is a

"fast" layer that connects $k$ RNN cells $F_1, \ldots, F_k$ in series; the other is a "slow" layer that consists of a single RNN cell $S$. The organization is roughly as follows: $F_1$ receives the input from the mini-batch and feeds its state into $S$; $S$ feeds its state into $F_2$; the output of $F_k$ is the probability distribution of the predicted character.

Table 3 outlines the performance of RUM and FS-RUM models along with other results in the PTB data set, in which we present the improved test BPC. Additionally, we test the performance of a simple RUM and a two-layer RUM. Comparing our FS-RUM-2 to the baseline FS-LSTM-2 (Mujika et al. (2017)) we improve the test evaluation by 0.001 BPC and the validation evaluation by 0.002 BPC. More detail are demonstrated in section D in the appendix.

| Model | BPC | # Parameters |
|---|---|---|
| Zoneout LSTM (Krueger et al. (2016)) | 1.27 | – |
| RUM 2000 (ours) | 1.28 | 8.9M |
| $2 \times$ RUM 1500 (ours) | 1.26 | 16.4M |
| HM-LSTM (Chung et al. (2016)) | 1.24 | – |
| HyperLSTM (Ha et al. (2016)) | 1.219 | 14.4M |
| NASCell (Zoph & V. Le (2016)) | 1.214 | 16.3M |
| FS-LSTM-4 (Mujika et al. (2017)) | 1.193 | 6.5M |
| FS-LSTM-2 (Mujika et al. (2017)) | 1.190 | 7.2M |
| FS-RUM-2 (ours) | **1.189** | 11.2M |

Table 3: With FS-RUM-2 we achieve the state-of-the-art test result on the Penn Treebank task.

FS-RUM-2 generalizes better than other gated models, such as GRU and LSTM, because it learns efficient patterns for activation in its kernels. Such a skill is useful for the large Penn Treebank data set, as with its special diagonal structure, the RUM cell in FS-RUM-2 activates the hidden state effectively. We discuss this representational advantage in section 5.1.

## 5 DISCUSSION

### 5.1 VISUAL ANALYSIS

One advantage of the Rotational Unit of Memory is that it allows the model to encode information in the phase of the hidden state. In order to demonstrate the structure behind such learning, we look at the kernels that generate the target memory $\tau$ in the RUM model. Figure 3 (a) is a visualization for the Recall task that demonstrates the diagonal structure of $W_{hh}^{(1)}$ which generates $\tau$ (a diagonal structure is also present $W_{hh}^{(2)}$, but it is contrasted less). One way to interpret the importance of the diagonal contrast is that each neuron in the hidden state plays an important role for learning since each element on the diagonal activates a distinct neuron.

Moreover, the diagonal structure is not task specific. For example, in Figure 3 (b) we observe a particular $W_{hh}^{(2)}$ for the target $\tau$ on the Penn Treebank task. The way we interpret the meaning of the diagonal structure, combined with the off-diagonal activations, is that probably they encode grammar and vocabulary, as well as the links between various components of language.

### 5.2 THEORETICAL ANALYSIS

It is natural to view the Rotational Unit of Memory and many other approaches using orthogonal matrices to fall into the category of *phase-encoding architectures*: $R = R(\boldsymbol{\theta})$, where $\boldsymbol{\theta}$ is a phase information matrix. For instance, we can parameterize any orthogonal matrix according to the Efficient Unitary Neural Networks (EUNN, Jing et al. (2017b)) architecture: $R = \prod_{i=0}^{N} U_0(\theta^i)$, where $U_0$ is a block diagonal matrix containing $N/2$ numbers of 2-by-2 rotations. The component $\theta_i$ is an one-by-$(N/2)$ parameter vector. Therefore, the rotational memory equation in our model can be represented as

$$R_t = \prod_{i=0}^{N} U_0(\theta_t^i) = \prod_{i=0}^{N} U_0(\theta_{t-1}^i) \cdot \prod_{i=0}^{N} U_0(\phi_t^i) \qquad (2)$$

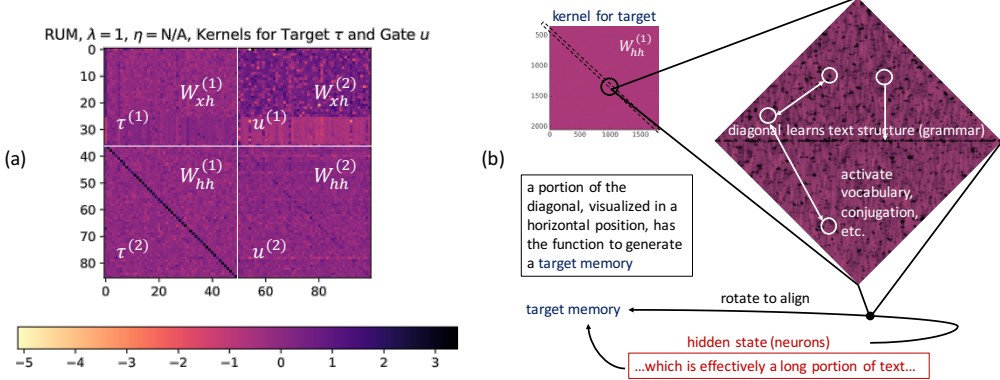

Figure 3: **The kernel generating the target memory for RUM is following a diagonal activation pattern, which signifies the sequential learning of the model**. (a) A temperature map of the values of the variables when the model is learned. The task is Associative Recall, $T = 50$, and the model is RUM, $\lambda = 1$, with $N_h = 50$ and without time normalization. (b) An interpretation of the function of the diagonal and off-diagonal activations of RUM's $W_{hh}$ kernel on NLP tasks. The task is Character Level Penn Treebank and the model is $\lambda = 0$ RUM, $N_h = 2000$, $\eta = 1.0$. See section E for additional examples.

where $\theta_t$ are rotational memory phase vectors at time $t$ and $\phi$ represents the phases generated by the operation Rotation correspondingly. Note that each element of the matrix multiplication $U_0(\theta^i) \cdot U_0(\phi^i)$ only depends on one element from $\theta^i$ and $\phi^i$ each. This means that, to cancel out one element $\theta^i$, the model only needs to learn to express $\phi^i$ as the negation of $\theta^i$.

As a result, our RNN implementation does not require a reset gate, as in GRU or GORU, because the forgetting mechanism is automatically embedded into the representation (2) of phase-encoding.

Thus, the concept of phase-encoding is simply a special sampling on manifolds generated by the special orthogonal Lie group $SO(N)$. Now, let $N = N_h$ be the hidden size. One way to extend the current RUM model is to allow for $\lambda$ to be any real number in the associative memory equation $R_t = (R_{t-1})^\lambda \cdot \text{Rotation}(\tilde{\varepsilon}_t, \boldsymbol{\tau}_t)$. This will expand the representational power of the rotational unit. The difficulty is to mathematically define the raising of a matrix to a real power, which is equivalent to defining a logarithm of a matrix. Again, rotations prove to be a natural choice since they are elements of $SO(N_h)$, and their logarithms correspond to elements of the vector space of the Lie algebra $\mathfrak{so}(N_h)$, associatied to $SO(N_h)$.

### 5.3 COMPARISON TO CAPSULE NET

We want to clarify that RUM and Capsule Net are not equivalent in terms of learning representations, but they share notable spiritual similarities.

**A parallel between RUMs state and Capsules representation**. The hidden state in our model is viewed as a vector in an Euclidean space $\mathbb{R}^n$ – it has an orientation and a magnitude. In a similar fashion, a capsule is a vector that has an orientation and a magnitude. Both RUM and Capsule Net learn to manipulate the orientation and magnitude of their respective components.

**The Rotation operation and the Routing mechanism**. Both mechanisms are ways of manipulating orientations and magnitudes. In the routing mechanism we start from priors (linearly generated from the input to the given layer of capsules), then generate outputs, and finally measure the dot product between the priors and the output. This dot product essentially measures the similarity between the two vectors through the cosine of the angle between them. This relative position between the two vectors is used for effective routing, so that the orientations of the capsules can be manipulated iteratively. For rotation mechanism, we start with the embedded input vector (an alternative of the priors) and then generate the target memory (an alternative of the outputs). Then we measure (encode) the rotation between the embedded input and the target memory (an alternative of taking

the dot product). And finally we use that encoded rotation to change the orientation of the hidden state (the iterative process of the routing mechanism).

**Main Difference.** The hidden state in RUM usually has a much larger dimensionality than the capsules that are used in Sabour et al. (2017). Hence, effectively, we demonstrate how to manipulate orientations and magnitudes of a much higher dimensionality (for example, we have experimented with hidden sizes of 1000 and 2000 for language modeling).

### 5.4 FUTURE WORK

For future work, the RUM model can be applied to other higher-level RNN structures. For instance, in section 4.4 we already showed how to successfully embed RUM into FS-RNN to achieve state-of-the-art results. Other examples may include Recurrent Highway Networks (Zilly et al. (2017)), HyperNetwork (Ha et al. (2016)) structures, etc. The fusion of RUM with such architectures could lead to more state-of-the-art results in sequential tasks.

## 6 CONCLUSION

We proposed a novel RNN architecture: Rotational Unit of Memory. The model takes advantage of the unitary and associative memory concepts. RUM outperforms many previous state-of-the-art models, including LSTM, GRU, GORU and NTM in synthetic benchmarks: Copying Memory and Associative Recall tasks. Additionally, RUM's performance in real-world tasks, such as question answering and language modeling, is competetive with that of advanced architectures, some of which include attention mechanisms. We claim the Rotational Unit of Memory can serve as the new benchmark model that absorbs all advantages of existing models in a scalable way. Indeed, the rotational operation can be applied to many other fields, not limited only to RNN, such as Convolutional and Generative Adversarial Neural Networks.

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

# APPENDIX

## A    COPYING MEMORY TASK

The alphabet of the input consists of symbols $\{a_i\}, i \in \{0, 1, \cdots, n-1, n, n+1\}$, the first $n$ of which represent data for copying, and the remaining two form "blank" and "marker" symbols, respectively. In our experiment $n = 8$ and the data for copying is the first 10 symbols of the input. The expectation from the RNN model is to output "blank" and, after the "marker" appears in the input, to output (copy) sequentially the initial data of 10 steps.

## B    ASSOCIATIVE RECALL TASK

The sequences for training are randomly generated, and consist of pairs of "character" and "number" elements. We set the key to always be a "character". We fix the size of the "character" set equal to half of the length of the sequence and the size of the "number" set equal to 10. Therefore, the total category has a size of $T/2 + 10 + 1$.

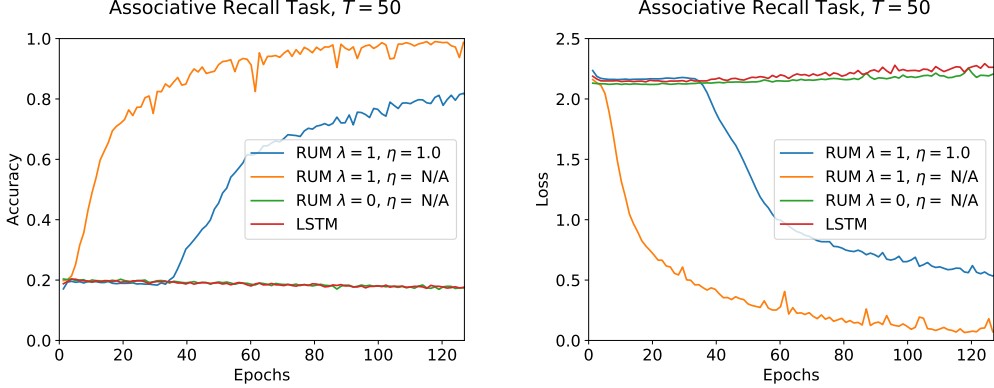

Figure 4: **The associative memory provided by rotational operation** Rotation **enables RUM to solve the Associative Recall Task**. The input sequences is 50 . For all models $N_h = 50$. For the training of all models we use RMSProp optimization with a learning rate of 0.001 and a decay rate of 0.9; the batch size $N_b$ is 128. We observe that it is necessary to tune in the associative memory via $\lambda = 1$ since $\lambda = 0$ RUM does not learn the task.

## C    QUESTION ANSWERING BABI TASK

In this task, we train 20 models jointly on each sub-task. All of them use a 10k data set, which is divided into 90% of training and 10% of validation. We first tokenize all the words in the data set and combine the story and question by simply concatenating two sequences. Different length

sequences are filled with "blank" at the beginning and the end. Words in the sequence are embedded into dense vectors and then fed into RNN in a sequential manner. The RNN model outputs the answer prediction at the end of the question through a softmax layer. We use batch size of 32 for all 20 subsets. The model is trained with Adam Optimizer with a learning rate 0.001. Each subset is trained with 20 epochs and no other regularization is applied.

| Task | RUM (ours) | LSTM (Weston et al.) | GORU (Jing et al.) | MemN2N (Sukhbaatar el al.) |
|---|---|---|---|---|
| Single Supporting Fact | 79.7 | 50 | 46 | 100 |
| Two Supporting Facts | 39.4 | 20 | 40 | 92 |
| Three Supporting Facts | 46.6 | 20 | 34 | 60 |
| Two Arg. Relations | 100 | 61 | 63 | 97 |
| Three Arg. Relations | 96.8 | 70 | 87 | 87 |
| Yes/No Questions | 84.7 | 48 | 54 | 94 |
| Counting | 89.1 | 49 | 78 | 83 |
| Lists/Sets | 74.0 | 45 | 75 | 90 |
| Simple Negation | 84.0 | 64 | 63 | 87 |
| Indefinite Knowledge | 75.7 | 44 | 45 | 85 |
| Basic Coreference | 90.6 | 72 | 69 | 99 |
| Conjunction | 95.0 | 74 | 70 | 99 |
| Compound Coref. | 91.5 | 94 | 93 | 99 |
| Time Reasoning | 43.9 | 27 | 38 | 98 |
| Basic Deduction | 58.8 | 21 | 55 | 100 |
| Basic Induction | 47.1 | 23 | 44 | 99 |
| Positional Reasoning | 60.3 | 51 | 60 | 49 |
| Size Reasoning | 98.5 | 52 | 91 | 89 |
| Path Finding | 10.2 | 8 | 9 | 17 |
| Agent's Motivations | 98.0 | 91 | 98 | 100 |
| Mean Performance | 73.2 | 49 | 60 | 86 |

Table 4: Question Answering task on bAbI dataset. Test accuracy (%) on LSTM, MemN2N, GORU and RUM. RUM significantly outperforms LSTM/GORU and has a performance close to that of MemoryNN, which uses an attention mechanism.

## D  CHARACTER LEVEL PENN TREEBANK TASK

For all RNN cells we apply layer normalization (Ba et al. (2016b)) to the cells and to the LSTM gates and RUM's update gate and target memory, zoneout (Krueger et al. (2016)) to the recurrent connections, and dropout (Srivastava et al. (2014)) to the FS-RNN. For training we use Adam optimization (Kingma & Ba (2014)). We apply gradient clipping with maximal norm of the gradients equal to 1.0. Table 5 lists the hyper-parameters we use for our models.

We embed the inputs into a higher-dimensional space. The output of each models passes through a softmax layer; then the probabilities are evaluated by a standard cross entropy loss function. The *bits-per-character* (BPC) loss is simply the cross entropy with a binary logarithm.

Table 7 outlines the performance of all variances of RUM models. Mujika et al. (2017) achieve their record with FS-LSTM-2, by setting $F_{1,2}$ and $S$ to LSTM. The authors in the same paper suggest that the "slow" cell has the function of capturing long-term dependencies from the data. Hence, it is natural to set $S$ to be a RUM, given its memorization advantages. In particular, we experiment with FS-RUM-2, for which $S$ is a RUM and $F_{1,2}$ are LSTM, as shown in figure 5. Additionally, we compare the validation performance of derivative models of our baseline FS-RUM-2 model in figure 6.

Empirically, we discovered that a time normalization $\eta = 1.0$ works best for RUM when gradient clipping norm is also 1.0.

## E  VISUALIZATION

| Hyper-parameters | RUM | $2 \times$ RUM | FS-RUM-2 |
|---|---|---|---|
| Non-recurrent dropout | 0.35 | 0.35 | 0.35 |
| Cell zoneout | 0.5 | 0.5 | 0.5 |
| Hidden zoneout | 0.1 | 0.1 | 0.1 |
| Fast cell size (LSTM) | N/A | N/A | 700 |
| Associative power $\lambda$ | 0 | 0 | 0 |
| Time normalization $\eta$ | 1.0 | 0.3 | 1.0 |
| Slow cell size (RUM) | 2000 | 1500 | 1000 |
| $T$ length | 150 | 150 | 150 |
| Mini-batch size | 128 | 128 | 128 |
| Input embedding size | 128 | 128 | 128 |
| Initial learning rate | 0.002 | 0.002 | 0.002 |
| Epochs | 100 | 100 | 360 |

Table 5: Hyper-parameters for the Character Level Penn Treebank Task.

| Learning rate | Epochs |
|---|---|
| 0.002 | 1-180 |
| 0.0001 | 181-240 |
| 0.00001 | 241-360 |

Table 6: Suggested learning rate pattern for training FS-RUM-2 with a standard Adam optimizer.

| Model | BPC | # Parameters |
|---|---|---|
| FS-LSTM-4 (Mujika et al. (2017)) | 1.193 | 6.5M |
| FS-LSTM-2 (Mujika et al. (2017)) | 1.190 | 7.2M |
| FS-RUM-2(B)+(S)1000-1000 (ours) | 1.194 | 17.6M |
| FS-RUM-2(B)+(S)1000-1200 (ours) | 1.193 | 19.9M |
| FS-RUM-2(B)+(S)900-1200 (ours) | 1.192 | 17.5M |
| FS-RUM-2(B)+LR (ours) | 1.190 | 11.2M |
| FS-RUM-2(B)+(S)800-1100 (ours) | **1.189** | 14.2M |
| FS-RUM-2 (ours) | **1.189** | 11.2M |

Table 7: FS-RUM-2(B)+LR is the baseline FS-RUM-2 except that the learning rate equals 0.003. FS-RUM-2(B)+(S)800-1100, 900-1200, 1000-1000 and 1000-1200 are derivative models of FS-RUM-2, which are defined in figure 6, and improve the validation performance of the baseline FS-LSTM-2 model. In figure 6 we also show the 0.002 improvement to the validation BPC loss, achieved by FS-RUM-2(B)+(S)800-1100.

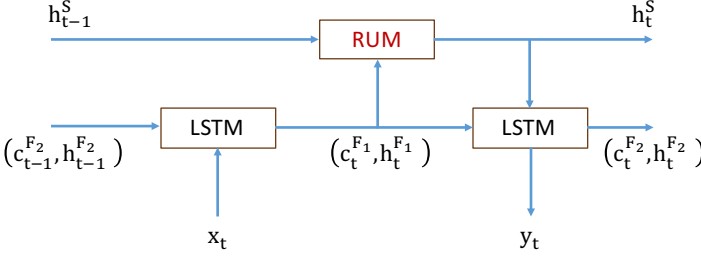

Figure 5: **The FS-RUM-2 high-level architecture**. In our experiments we combine a slow RUM cell with fast LSTM cells in the FS-RNN model.

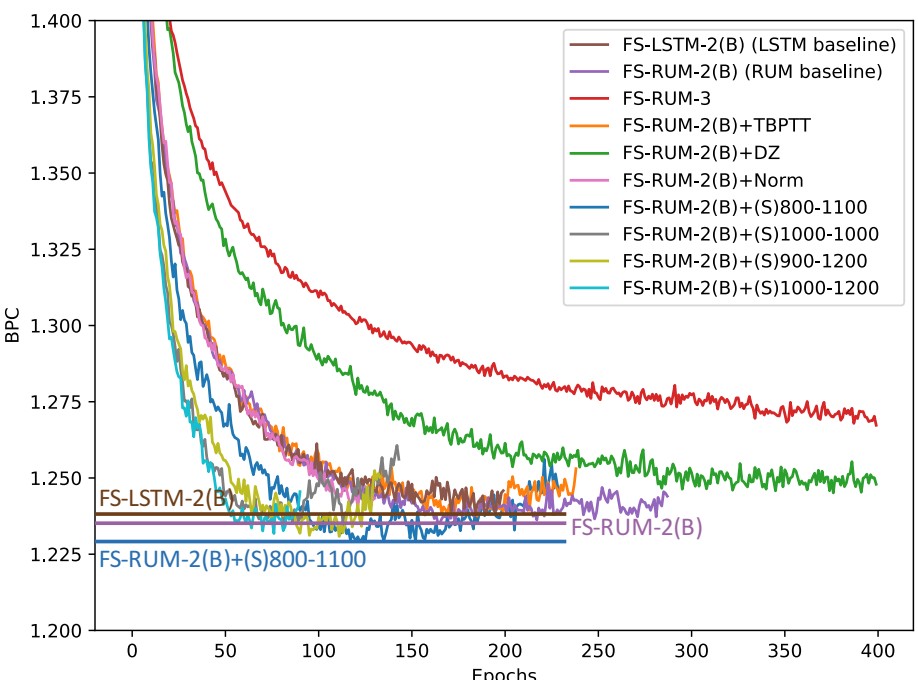

Figure 6: **Examples of validation BPC vs Epochs for each model in the PTB task.** The FS-LSTM-2(B) model is the previous state-of-the-art by Mujika et al. (2017), which we take as an LSTM baseline. The FS-RUM-2(B) model is the FS-RUM-2 model, descibred in table 5, which serves as our RUM baseline. FS-RUM-3 has the same hyper-parameters as the baseline FS-RUM-2(B), except that it has 3 fast cells LSTM, each with a hidden state of 500. FS-RUM-2(B)+TBPTT is the same as FS-RUM-2(B) except that the $T$ length is 200. FS-RUM-2(B)+DZ is the same as FS-RUM-2(B) except that the (non-recurrent dropout, cell zoneout, hidden zoneout) is (0.4, 0.5, 0.2). FS-RUM-2(B)+Norm is the same as FS-RUM-2(B) except that the time normalization norm $\eta$ is 1.3. FS-RUM-2(B)+(S)800-1100 is the same as FS-RUM-2(B) except that the fast cells' size is 800 and the slow cell's size is 1100. FS-RUM-2(B)+(S)1000-1000 is the same as FS-RUM-2(B) except that the fast cells' size is 1000 and the slow cell's size is 1000. FS-RUM-2(B)+(S)900-1200 is the same as FS-RUM-2(B) except that the fast cells' size is 900 and the slow cell's size is 1200.

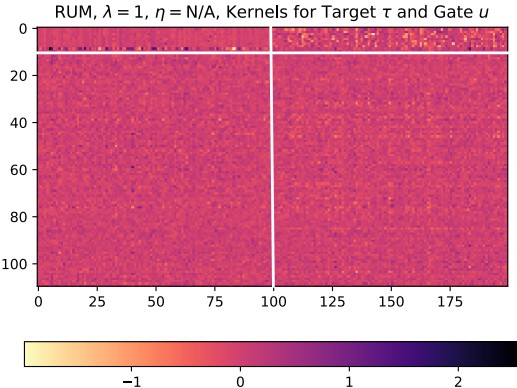

Figure 7: The collection of kernels for $\lambda = 1$ RUM, $N_h = 100$, $\eta = $ N/A for the Copying task, $T = 500$.

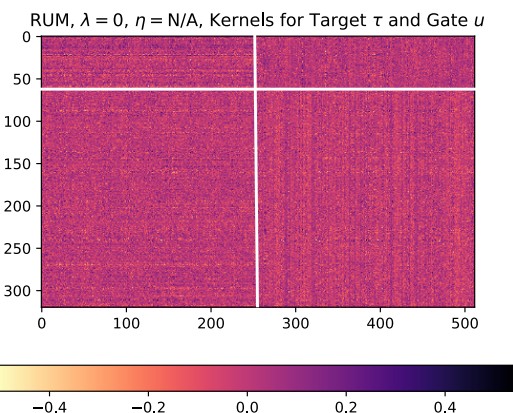

Figure 8: The collection of kernels for $\lambda = 0$ RUM, $N_h = 256$, $\eta = $ N/A for the Question Answering bAbI Task.

