# OpenReview forum: "Rotational Unit of Memory "
_ICLR.cc/2018/Conference — Invite to Workshop Track_

### Official Review · AnonReviewer2 · 2017-11-27

**Rating:** 4
**Confidence:** 4

**Review:**

The authors of this paper propose a new type of RNN architecture that modifies the reset gate of GRU with a rotational operator, where this rotational operator serves as an associative memory of their RNN model. The idea is sound, and the way they conduct experiments also make sense. The motivation and the details of the rotational memory are explained clearly. However, the experimental results reported in the paper seem to be a bit weak to support the claims made by the authors.

The performance improvements are not so clear to me. Especially, in the character level language modeling, the BPC improvement is only 0.001 when choosing the SOTA model of this dataset as the base architecture. The test BPC score is obtained as a single-run experiment on the PTB dataset, and the improvement seems to be too small. In the copying memory task shown in Section 4.1, how did GORU performed when T=200?

On the Q&A task, using the bAbI set (Section 4.3), RUM is said to be *significantly outperforming* GORU when the performance gap is 13.2%, and then, it is also said that RUM’s performance *is close to* the MeMN2N when the performance gap is 12.8%. Both performance gaps seem to be very close to each other, but the way they are interpreted in the paper is not.

Overall, the writing is clear, and the idea sounds interesting, but the experimental results are not strongly correlated with the claims made in the paper. In the visual analysis, the authors assume that RUM architecture might be the architecture that utilizes the full representational power of models like RNNs. If this is the case, I would expect to see more impressive improvements in the performance, assuming that all the other conditions are properly controlled.

I would suggest evaluating the model on more datasets.

Minor comments:
In Section 2.2: Hopflied -> Hopfield
In Section 3.2: I believe the dimension of b_t should be 2*N_h

---

> ### Author Response · Authors · 2017-12-05
> **Re: Review**
>
> Dear Reviewer,
>
> We thank you for the constructive review! For evaluation of RUM we wanted to test the model on diverse benchmark tasks, ranging from the Copying Memory Task and Character-level Language Modeling on PTB, which require a varied set of skills, including long-term memory capacity, associative skill, short-term forgetting mechanisms, etc. Our confidence in RUM is motivated by the state-of-the-art-like performance of the model in all those tasks.
>
> Thank you for suggesting to expand the experimental section. As far as the current results are concerned, we are working on a finer grid search, which can yield more impressive improvements. We are also evaluating RUM on a larger data set--enwik8: it is possible these simulations will not finish in time (before the deadline), but we’ll try.
>
> We agree with your comment on the Q&A task, and we will rephrase this part of the experimental discussion. However, we want to explain why our result in this task is strong. Attention mechanism models hold the record for all Q&A tasks nowadays. Nevertheless, RNN models are still more responsible for long-term memory which should improve the SOTA when combined with attention mechanisms. Frankly, there is still a lack of studies on combining novel RNNs with attention mechanisms to achieve SOTA. Thus, this should not prevent studies on better fundamental RNN models, e.g. Cooijmans et al (2016). For future work, we plan to apply RUM to other Q&A data sets.
>
> Finally, GORU learns the Copying Memory Task for T=200; we will update our figure. We will also implement your minor comments and update the paper accordingly.
>
> Thank you!
>
> References:
>
> Tim Cooijmans, Nicolas Ballas, César Laurent, Çaglar Gülçehre & Aaron Courville. Recurrent Batch Normalization. ICLR 2017 arXiv preprint arXiv:1603.09025, 2016.

---

> > ### Author Response · Authors · 2018-01-05
> > **Performance of RUM**
> >
> > Dear Reviewer,
> >
> > We are currently evaluating RUM on larger data sets. For example, right now we work on the enwik8 data set. So far, we can run models up to 21M parameters. FS-RUM-2 with 21M params. achieves 1.37 BPC for 35 epochs on enwik8. This is not the state-of-the-art, but note that the state-of-the-art is a much larger model that has 94M params. (Mujika et al. (2017)). We are currently working on fitting larger RUM model to our hardware, so that we can test the RUM model of size about 100M params.
> >
> > Moreover, the state-of-the-art models on enwik8 (Mujika et al. (2017)) are simply larger versions of the previous state-of-the-art models on the PTB task (Mujika et al. (2017)). We show that a RUM version of those models slightly improves the test by 0.001 and validation by 0.002; therefore we think that PTB is a particularly important data set to experiment with. Moreover, we give a positive answer to the suggestion by Mujika et al. (2017) in their conclusion that the slow LSTM cell may be replaced with a cell that has a better long term memory capacity. As we demonstrate, RUM is suitable for such replacements--a feat, which we think gives RUM scaling power. We hope that our additional experiments provide a clearer picture of the performance of the RUM model.
> >
> > In our analysis we do not exactly assume that RUM gives full representational power. We simply conduct a visual analysis of the kernels and offer our interpretation of the results. Whether or not RUM gives full representational power is very difficult to be evaluated, as it is mostly task specific; moreover, such a statement is not a necessary assumption for any part of the paper, thus we agree to rephrase that part of our visual analysis, so that our interpretation is clearer.
> >
> > On that note, we also think that currently it is difficult to judge whether an improvement on a particular task, such as the PTB character level, is weak or not. There has been much research on PTB already, and we may not know where the performance limit of deep learning models stands. For this reason, to incorporate your concerns, we have rephrased some of our statements in the paper.
> >
> > Best wishes!
> >
> > References
> >
> > Asier Mujika, Florian Meier, and Angelika Steger. Fast-slow recurrent neural networks. NIPS arXiv preprint arXiv:1705.08639, 2017.

---

### Official Review · AnonReviewer3 · 2017-11-28
**Nice formulation of phase-coding memory for RNNs**

**Rating:** 6
**Confidence:** 3

**Review:**

The paper proposes a RNN memory cell updating using an orthogonal rotation operator. This approach falls into the phase-encoding architectures. Overall the author's idea of generating a rotation operator using the embedded input and the transformed hidden state at the previous step is clever. Modelling this way makes the 'generating matrix' W_hh learn to couple the input to the hidden state (which contain information in the past) via the Rotation operator.

I have several concerns:

- The author should discuss the intuition why the rotation has to be from the generated memory target τ to the embeded input ε but not the other way around or other direction in this 2D subspace.
- The description of parameter meter τ is not clear. Perhaps the author meant τ is the generated parameter via the parameter matrix W_hh acting upon the hidden state h_{t-1}
- The idea of evolving the hidden state by an orthogonal matrix, of which the rotation is a special case, is similar to the GORU paper, which directly parametrizes the 'rotation' matrix. Therefore I am wondering if the better performance of this work than the GORU is because of the difference in parameterization or by limiting the orthogonal transform to only rotations (hence modelling only the phase of the hidden state). Perhaps an additional experiment is needed to verify this.

---

> ### Author Response · Authors · 2017-12-05
> **Re: Nice formulation of phase-coding memory for RNNs**
>
> Dear Reviewer,
>
> We thank you for the thoughtful review! We believe that your question about the choice of the initial and final vectors, encoding the rotation, within the 2D subspace can lead to new interesting results. We might introduce two new parameters—alpha and beta—that define the rotation from the embedded input to a linear combination alpha*u+beta*v, where u and v form an orthonormal basis of the 2D subspace. The coefficients alpha and beta can be learned by backpropagation.
>
> Currently, we rotate from the embedded input to the target vector; if we decide to flip the encoding (from target to embedded input) we expect to obtain comparable results to the current ones since we only reverse the orientation of the rotation.
>
> We thank you for your comments on the description of tau and will update the discussion accordingly. We will also conduct an additional experiment that will answer your questions about the comparison between RUM and GORU.
>
> Thank you!

---

> > ### Author Response · Authors · 2017-12-31
> > **On the choice of the particular direction in the 2D subspace**
> >
> >
> > Dear Reviewer,
> >
> > We finished our experiments on trying different linear combinations within the 2D subspace and reversing the orientation of the rotation, and concluded that the performance stays the same as the current model, which aligned with our original intuition. Here we will motivate the outcome of our experiment through a simple observation.
> >
> > We want the rotation to depend on the input and the hidden state. The simplest way for this is to encode the rotation from the embedded input to a linear combination of the input and the hidden state. This linear combination can be written as alpha * x_emb + beta * target_memory. The magnitude of the vectors forming the rotation does not matter. Thus, we can divide by beta to get alpha/beta * x_emb + target_memory, i.e. only alpha/beta matters. Since the magnitude of x_emb is determined by weights/bias, the degree of freedom of alpha/beta is absorbed into the weights/bias. In practice, rotating to the target_memory solely or to x_emb + target_memory gives the same performance.
> >
> > Hopefully this helps to answer your question.
> >
> > Thank you!

---

> > > ### Author Response · Authors · 2018-01-05
> > > **Remarks about RUM and GORU**
> > >
> > > Dear Reviewer,
> > >
> > > We agree that the GORU model directly parameterizes the rotation matrix. For easier tasks like the copying task, fixed parametrization may have advantages. However, in most tasks, the “learnware” structure that RUM provides (Balduzzi et al. (2016)), performs better thanks to its flexibility of adjusting weight matrices in RNN, which require different abilities in different time steps. In addition, the learnware structure of RUM has the same representative ability as GORU in terms of the “rotation matrix”. Both of them are able to occupy full orthogonal space (more accurately, orthogonal matrix manifolds with all positive eigenvalues). This assures that RUM will perform no worse than GORU even though it might learn slower in some tasks. In other tasks, such as language modeling, RUM is better because of its learnware structure, as demonstrated by experiments in the original GORU paper (Jing et al. (2017)) vs. experiments in our paper.
> > >
> > > Best wishes!
> > >
> > > References
> > >
> > > David Balduzzi and Muhammad Ghifary. Strongly-Typed Recurrent Neural Networks. Proceeding ICML'16 Proceedings of the 33rd International Conference on Machine Learning. 48. 1292-1300, 2016
> > >
> > > Li Jing, Caglar Gulcehre, John Peurifoy, Yichen Shen, Max Tegmark, Marin Soljacic, and Yoshua Bengio. Gated orthogonal recurrent units: On learning to forget. arXiv preprint arXiv:1706.02761, 2017a.

---

### Official Review · AnonReviewer1 · 2017-11-28
**On Rotational Unit of Memory**

**Rating:** 5
**Confidence:** 4

**Review:**

Summary:
This paper proposes a way to incorporate rotation memories into gated RNNs. They use a specific parametrization of the rotation matrices. They run experiments on several toy tasks and on language modelling with PTB character-level language modeling (which I would still consider to be toyish.)


Question:
Can the rotation proposed here cause unintentional forgetting by interleaving the memories? Because in some sense rotations are glorified summation in high-dimensions, if you do a full-rotation of a vector (360 degrees) you can end up in the same location. Thus the model might overwrite into its past memories.

Pros:
Proposes an interesting way to incorporate the rotation operations into the gated architectures.

Cons:
The specific choice of rotation operation is not very well justified.
This paper more or less uses the same architecture from Jing et al 2017 from EU-RNNs with a different parametrization for the rotation matrices.
The experiments are still limited to simple small-scale tasks.


General Comments:

The idea and the premise of this paper is interesting. In general the paper seems to be well-written. However the most important part of the paper section 3.1 is not very well justified. Why this particular parameterization of the rotation matrices is used and where does actually that come from? Can you point out to some citation? I think the RUM architecture section also requires better explanation on for instance why why R_t is parameterized that way (as a multiplicative function of R_{t-1}). A detailed ablation study would help too.

The model seems to perform really close to the GORU on Copying Task. I would be interested in seeing comparisons to GORU on “Associative Recall” as well. On QA task, which subset of bAbI dataset have you used? 1k or 10k training sets?

On language modelling there is only insignificant difference between the FS-LSTM-2 with FS-RUM model. This does not tell us much.

---

> ### Author Response · Authors · 2017-12-05
> **Re: On Rotational Unit of Memory**
>
> Dear Reviewer,
>
> We thank you for the insightful review! We believe that our concept of using rotation memories can be used in a large set of deep learning models, including RNNs. Our paper serves to introduce a particular construction (Rotation) that realizes the concept of rotation memories, and then to illustrate advantages of that construction by modifying gated models (RUM).
>
> We agree that testing RUM on tasks with larger data sets would bolster the case for RUM. Currently we are running our model on enwik8: it is possible our simulations will not finish in time (before the deadline), but we’ll try.
>
> We believe that your comment about “unintentional forgetting” is interesting and will investigate it further. The RUM model utilizes rotations defined by projections into different (in general) 2d planes (defined by the embedded input vector and the target vector) under which going back to the same point unintentionally (after making a cycle of 360 degrees) is unlikely. Another way to think about this is by viewing the rotations as rotating a unit vector on an (N_h-1)-sphere, where N_h is the hidden size. Since N_h is typically not small, the probability of ending at the same point after a full cycle is negligible.
>
> While RUM is only partially motivated by GORU, the RUM model introduces two crucial new concepts, which, we believe, substantially bolster its performance compared to GORU, and many other approaches: 1. The rotation operation is not parameterized directly, as in GORU, but instead it is extracted from the new input and the previous hidden state. In this sense, to parallel our model with the literature,  RUM is a “firmware” structure instead of a “learnware” structure as discussed in Balduzzi et al. (2016): our rotation does not require additional parameters to be defined. 2. RUM has an associative memory structure, which is not present in GORU, and more importantly, it is vital for the learning of the Associative Recall task (soon we will report on the inability of GORU to learn the task for T=30 and 50; note that RUM succeeds for T=30 and 50). Moreover, the multiplicative recursive definition of R_t is required to maintain an orthogonal matrix and have an interpretation of phase accumulation because of the multiplicative nature of rotations. We believe that this is the first example of a multiplicative function used for associative memory, contrasting the recursions in Ba et al (2016) and  Zhang et al (2017).
>
> As far as rotations are concerned, they are key objects in a variety of fields such as quantum physics and  the theory of Lie groups. If one wants to find inspirations for constructions, similar to ours in section 3.1., they could consult standard books on those subjects (Sakurai et. al (2010), Artin (2011)). This particular parameterization for the rotation is a natural way to define a differentiable orthogonal operation within the RNN cell. Other ways to extract an orthogonal operation from elements in the RNN cell are still possible. Some approaches are as follows: 1. Use a skew-symmetric matrix A to define the orthogonal operator e^A; 2. Use a permutation operator. However, those constructions are difficult to implement and do not offer a natural intuition about encoding memory. We recognize that other constructions are also feasible and potentially interesting for research; however, we believe that our construction of the Rotation is simple and offers enough intuition (and results) to spur more research in constructing  successful models other than RUM. We will update the discussion about the motivation of the rotational memory accordingly, but we will leave other constructions as a topic for further work (i.e. for another conference).
>
> Finally, we used the 10k training set for the QA task.
>
> Thank you!
>
> References:
>
> Jimmy Ba, Geoffrey Hinton, Volodymyr Mnih, Joel Z. Leibo, Catalin Ionescu. Using Fast Weights to Attend to the Recent Past. arXiv preprint arXiv:1610.06258, 2016.
>
> Wei Zhang and Bowen Zhou. Learning to update auto-associative memory in recurrent neural networks for improving sequence memorization. arXiv preprint arXiv:1709.06493, 2017.
>
> David Balduzzi and Muhammad Ghifary. Strongly-Typed Recurrent Neural Networks. Proceeding ICML'16 Proceedings of the 33rd International Conference on International Conference on Machine Learning. 48. 1292-1300, 2016.
>
> J. J. Sakurai and Jim J. Napolitano. Modern Quantum Mechanics (2nd edition). Pearson, 2010.
>
> Michael Artin. Algebra (2nd edition). Pearson, 2011.

---

> > ### Author Response · Authors · 2018-01-05
> > **On the motivation and scalability of RUM**
> >
> > Dear Reviewer,
> >
> > Inspired by your comments, we worked on clarifying the motivation for RUM and demonstrating signs of its potential applications to more complicated models and tasks.
> >
> > First of all, we believe that using rotations within neural networks is a natural choice, which is backed up by the mathematical properties of those operations. To us, rotations stand out, because through a simple construction, they combine a variety of concepts, important to deep learning, such as: gradient explosion/vanishing, associative memory, and also relate to novel propositions of deep learning models, such as capsules. We think that this inherent universality of the rotation operations helps RUM perform well on a diverse choice of tasks, each requiring a special learning capacity. Thus, we tested RUM on the copying task, question answering, associative recall and language modeling--4 conceptually diverse tasks.
> >
> > Second of all, because of the initial success of RUM on those 4 tasks, we are currently evaluating RUM on larger data sets. For example, right now we work on the enwik8 data set. So far, we can run models up to 21M parameters. FS-RUM-2 with 21M params. achieves 1.37 BPC for 35 epochs on enwik8. This is not the state-of-the-art, but note that the state-of-the-art is a much larger model that has 94M params. (Mujika et al. (2017)). We are currently working on fitting larger RUM model to our hardware, so that we can test the RUM model of size about 100M params.
> >
> > Finally, the state-of-the-art models on enwik8 (Mujika et al. (2017)) are simply larger versions of the previous state-of-the-art models on the PTB task (Mujika et al. (2017)). We show that a RUM version of those models slightly improves the test by 0.001 and validation by 0.002; therefore we think that PTB is a particularly important data set to experiment with. Moreover, we give a positive answer to the suggestion by Mujika et al. (2017) in their conclusion that the slow LSTM cell may be replaced with a cell that has a better long term memory capacity. As we demonstrate, RUM is suitable for such replacements--a feat, which we think gives RUM scaling power.
> >
> > Best wishes!
> >
> > References
> >
> > Asier Mujika, Florian Meier, and Angelika Steger. Fast-slow recurrent neural networks. NIPS arXiv preprint arXiv:1705.08639, 2017.

---

### Author Response · Authors · 2017-12-05
**Rotational memory constructions are similar concepts to Capsule models**

Dear Reviewers,

We thank you for your comments and suggestions! Our impression is that you acknowledge the novelty and potential of our rotational memory constructions. Hopefully, through the upcoming discussions we will reach to a clear understanding of the theoretical importance and the experimental promise of the Rotation operation and the RUM model. For example, recently we discovered that our paper and Sabour et al. (2017) have a similar conceptual background.

Best wishes!

Sara Sabour, Nicholas Frosst, Geoffrey Hinton. Dynamic Routing Between Capsules. NIPS 2017 (to appear) arXiv preprint arXiv:1710.09829, 2016.

---

> ### Public Comment · (anonymous) · 2017-12-14
> **Some questions on Rotational Unit of Memory**
>
> I am interested in the rotation operation and the rum architecture and also the capsule net.
> But I do not find the spiritual similarity between these two works, could you help specifying?
>
> Besides, there are some questions.
> 1. I am not sure why the rotation operator in RUM is to rotate the previous hidden state from embedded input direction to target vector direction and than plus the embedded input?
> 2. What is the purpose of the accumulation of rotation?
>
> Thank you!

---

> > ### Author Response · Authors · 2017-12-22
> > **A more detailed comparison between RUM and Capsules**
> >
> > Dear Reader,
> >
> > We thank you for your interest in the topic and questions! First we describe the spiritual similarity between our contributions and the concept of Capsules, presented in Sabour et al. (2017):
> >
> >
> > a. *A parallel between RUM’s state and Capsule’s representation*. Think about the hidden state in our model as a vector in the Euclidean space R^n -- it has an orientation and a magnitude. In a similar fashion, a capsule is a vector that has an orientation and a magnitude. Both RUM and Capsule Net learn to manipulate the orientation and magnitude of their respective components.
> >
> > b. *The Rotation operation and the Routing mechanism*. Both mechanisms are ways of manipulating orientations and magnitudes. In the routing mechanism we start from priors (linearly generated from the input to the given layer of capsules), then generate outputs, and finally measure the dot product between the priors and the output. This dot product essentially measures the similarity between the two vectors through the cosine of the angle between them. This relative position between the two vectors is used for effective routing, so that the orientations of the capsules can be manipulated iteratively. Now, compare this with the Rotation mechanism. We start with the embedded input vector (think about this as an alternative of the priors) and then generate the target memory (think about this as an alternative of the outputs). Then we measure (encode) the rotation between the embedded input and the target memory (think about this as an alternative of taking the dot product). And finally we use that encoded rotation to change the orientation of the hidden state (think about this as the iterative process of the routing mechanism).
> >
> > c. Some additional remarks: of course, RUM and Capsule Net are not equivalent models in terms of their learning representations, but they share notable spiritual similarities, as noted in a. and b. Note that the hidden state usually has a much larger dimensionality than the capsules that are used in Sabour et al. (2017). Hence, effectively, we demonstrate how to manipulate orientations and magnitudes of a much higher dimensionality (for example, we have experimented with hidden sizes of 1000 and 2000 for language modeling).
> >
> > To answer your question 1. Our main idea is to utilize the orientation of the hidden state, viewed as a vector in the Euclidean space R^n. Since we use RNNs, the way for this update to happen is to take an input and a previous hidden state, and then produce a new hidden state. We want the change of the orientation of the hidden state to be guided by the inputs. This is the reason why we compute the rotation between the embedded input and the target memory, because note that the target memory is essentially generated by the previous hidden state. So after we compute this “coupling” between the input and the previous hidden state, we can create a new hidden state. The way for this to happen is by rotating the old hidden state to get a new hidden state (with  a different orientation).
> >
> > From here, to answer your question 2. We got the natural idea of phase accumulation. By providing a phase accumulation property for the rotation operation we invoke an associative memory within our RUM cell. This multiplicative structure, with no previously explored analogues to the best of our knowledge, allows for a more flexible rotational manipulation of the hidden state (similar to the “routing mechanism” for the RUM cell). We were positively surprised to see that we can achieve the state-of-the-art on the Associative Recall task with significantly less parameters. The potential for exploring these accumulations of rotations are vast, as we note in the Conclusion section of our paper.
> >
> > Best wishes!
> >
> > References
> >
> > Sara Sabour, Nicholas Frosst, Geoffrey Hinton. Dynamic Routing Between Capsules.. NIPS 2017 arXiv preprint arXiv:1710.09829, 2016.

---

### Author Response · Authors · 2018-01-05
**Important updates in the paper**

Dear Readers,

We took the comments of the reviewers into serious consideration, and, along with the minor corrections, we implemented the following major changes in the paper:

1. We added the discussion about RUM and Capsules (sections 2.3 and 5.3).
2. We clarified the motivation behind using rotations for more efficient models, and some of the steps of the RUM construction in section 3.
3. We updated the experimental sections for all 4 of the considered tasks to include more comparisons with the GORU and EUNN baseline models.
4. We now present a more comprehensive language modeling experiments for the PTB data set in section 4.4. and also appendix D. We managed to get to a 0.001 BPC test improvement and a 0.002 BPC validation improvement.

To sum up, with this paper we want to demonstrate a new phase-encoding learning representation, give intuition about its efficiency by analysing it through the lens of well-established deep learning models, and then demonstrate that the proposed model achieved satisfactory performance, which improves the state-of-the-art slightly, in a diverse range of tasks.

Thank you for the constructive discussion!

---

### Decision · Program_Chairs · 2018-01-29
**ICLR 2018 Conference Acceptance Decision**

**Decision:**

Invite to Workshop Track

**Comment:**

although the authors argue that their experiments were selected from the earlier work from which major comparing approaches were taken, the reviewers found the empirical result to be weak. why not some real tasks (i do not believe bAbI nor PTB could be considered real) that could clearly reveal the superiority of the proposed unit against existing ones?